# Endothelial cell specific molecule 1 promotes epithelial-mesenchymal transition of cervical cancer via the E-box binding homeobox 1

Jie Qi[1,2], Jie Li[2], Xiaoyan Zhu[3], Sufen Zhao[1] *

**1** Department of Gynecology, The Second Hospital of Hebei Medical University, Shijiazhuang, Hebei, People's Republic of China, **2** Department of Gynecology, Hebei General Hospital, Shijiazhuang, Hebei, People's Republic of China, **3** Department of Gynecologic Oncology, Jilin Cancer Hospital, Chaoyang District, Changchun, Jilin, People's Republic of China

* jieqi1540@163.com

## Abstract

### Objective

To investigate the mechanism of endothelial cell specific molecule 1 (ESM1) promoting cervical cancer cell proliferation and EMT characteristics through zinc finger E-box binding homeobox 1 (ZEB1)/EMT pathway.

### Methods

The correlation between ESM1 expression and prognosis of cervical cancer patients was analyzed by bioinformatics. SiHa, HeLa cell lines and corresponding control cell lines with stable ESM1 expression were obtained. Cell proliferation ability was detected by CCK-8 assay. The invasion and migration ability of Hela and SiHa cells were detected by Transwell assay and scratch closure assay. Expressions of EMT-related markers E-cadherin and Vimentin were detected by real-time PCR. The ability of silenced ESM1 to tumor formation in vivo was detected by tumor formation in nude mice. The effects of aloe-emodin on inhibit ESM1 expression and its inhibitory effect on cervical cancer cells in vitro and in vivo were analyzed by the same method.

### Results

ESM1 was highly expressed in cervical cancer, and the high expression of ESM1 was associated with poor prognosis of cervical cancer patients. CCK-8 results showed that the proliferation, invasion and migration of Hela and SiHa cells were significantly reduced after siRNA interfered with ESM1 expression. Overexpression of ESM1 promoted the proliferation and migration of cervical cancer cells. Mechanism studies have shown that the oncogenic effect of ESM1 is realized through the ZEB1/PI3K/AKT pathway. High throughput drug screening found that aloe-emodin can target ESM1. Inhibitory effect of aloe emodin on ESM1/ZEB1/EMT signaling pathway and cervical cancer cells.

**Data Availability Statement:** "The data used and/or analyzed in the present study are available within the manuscript itself."

**Funding:** The author(s) received no specific funding for this work.

**Competing interests:** The authors have declared that no competing interests exist.

## Conclusion

The silencing of ESM1 expression may inhibit the proliferation, invasion, metastasis and epithelial-mesenchymal transformation of cervical cancer cells by inhibiting ZEB1/PI3K/AKT. Aloe-emodin is a potential treatment for cervical cancer, which can play an anti-tumor role by inhibiting ESM1/ZEB1.

## Introduction

Cervical cancer ranks as the fourth most commonly diagnosed cancer and one of the frequently leading causes of cancer-related death in women worldwide [1]. Although the early detected cervical cancer could be treated with a curable disease for surgical resection and concurrent chemoradiation, cervical cancer remains a major public problem even in developed countries for limited treatment with advanced cervical cancer [2, 3]. Distant metastasis is a key property of the advanced cervical cancer, the malignant progression mechanism of cervical cancer needs further explanation to provide improved therapeutic options.

Growing evidence demonstrated that epithelial-mesenchymal transition (EMT) played a key role in tumor malignant progression. EMT is a process in which the epithelial tumor cells lose polarity and cell-cell adhesion characters and gain metastasis and invasive properties [4, 5]. During the EMT process, the EMT-activator ZEB1, a member of the zinc finger transcription factor family, transcriptionally regulates the expression of epithelial/mesenchymal markers (such as E-cadherin and Vimentin) to promote the EMT progression [6, 7]. Upregulation of ZEB1 had been found in various tumors and a positively correlation between ZEB1 expression and tumor malignant progression was verified [8, 9]. Thus, a protein co-expressed with ZEB1 is likely to regulate the tumor cells through EMT process.

Endothelium-specific molecule 1 (ESM-1) was originally a 2kb cDNA sequence accidentally discovered during cloning of human umbilical vein endothelial cell cDNA library. This sequence can encode a soluble proteoglycan, also known as Endocan. The biological function of ESM-1 is still under investigation. Preliminary research results show that it has an important relationship with arteriosclerosis, inflammation, tumor development, the formation of blood vessels and lymphatic vessels, as well as a variety of biomolecular regulation mechanisms. Studies have found that ESM-1 is highly expressed in many human solid tumors such as lung cancer, breast cancer, uterine cancer and kidney cancer. At the same time, it was also found that the high expression of this factor is related to the prognosis of the tumor, metastasis, vascular hyperplasia and other factors [10, 11].

At present, although a lot of anti-tumor drugs have been appearing, the efficacy and price of these drugs are still not satisfactory. ESM-1 is significantly associated with important factors of tumor occurrence and development, such as chronic inflammation, tumor cells themselves, angiogenesis, lymphangiogenesis, etc. Inhibition of ESM-1 expression can inhibit the occurrence and development of tumors from multiple aspects. Therefore, the development of ESM-1 targeting drugs may open up a new chapter in cancer therapy. Aloe emodin (AE), a bioactive anthraquinone compound extracted from aloe and rhubarb, has a variety of pharmacological activities [12]. Aloe emodin has significant anti-tumor effects on various tumor cells, and its mechanism involves many ways, including cell cycle arrest, cell apoptosis induction and antitumor effects as an adjuvant [13–15]. However, whether AE can inhibit ESM1 expression has not been reported in the literature.

In this study, we found that ESM1 was significantly upregulated in cervical cancer, ESM1 was reported to regulate tumor progression in many cancer types including non-small-cell lung cancer, ovarian cancer, gastric cancer, and hepatocellular carcinoma [16–19]. However, whether ESM1 is related to the malignant progression and could mediate the EMT process in cervical cancer remain elusive, which needs for a comprehensive investigation. Meanwhile, we will discover potential target drugs based on the clarified mechanism, which will provide potential therapeutic strategy for clinical treatment of cervical cancer.

## Methods

### Bioinformatics analysis

The human protein atlas platform, LinkedOmics database, GEPIA database, and the GEO database were used to analyze the relationship between ESM1 gene and cervical cancer. TCGA (the cancer genome atlas home page) is a public database contains a variety of human tumor cancer genome change (http://cancergenome.nih.gov/abouttcga). To determine the expression level of ESM1 in cervical cancer, the TCGA database was used to analyze the expression level of ESM1 in cervical cancer and normal tissues. The GEPIA database (http://gepia.cancer-pku. cn/in-dex.html) is a gene expression analysis database of 9736 tumor and 8587 normal samples from the TCGA and GTEx databases developed by the team of Peking University. In this paper, GEPIA database and a GEO data of GSE527 downloaded from the GEO database were used to analyze the relationship between ESM1 gene and prognosis of cervical cancer patients. Single cell sequencing data was downloaded from The Human Protein Atlas (http://www. proteinatlas.org/).

### Protein interaction network analysis

The STRING (https://string-db.org/) database is an online analysis site for predicting protein interactions by calculating their latent functions [20]. The STRING database was used to analyze the interaction between ESM1 and co-expressed positive correlated proteins, so as to predict its role in cervical cancer EMT.

### Functional enrichment analysis

The ESM1 co-expression related genes were input into the STRING platform, and the GO functional enrichment analysis and KEGG pathway enrichment analysis of related genes were performed respectively. Enrichment biological functions and signaling pathways were screened according to P-value < 0.05.

### Cell culture and transfection

Cervical cancer cell lines SiHa and HeLa were purchased from the American Type Culture Collection (ATCC). Hela and SiHa cells were cultured in DMEM medium containing 10% fetal bovine serum in a 5% $CO_2$ incubator at 37˚C. ESM1 overexpression and GFP control plasmids were transfected into SiHa cells by lipid method. The operation steps were carried out in strict accordance with the instructions of Lipofectamine 2000. Cells were seeded in 6-well plates at 50% density for overnight cultures. ESM1 plasmid transfection group: add 4.0 μg negative control plasmid and 10 μL transfection reagent to each well of the negative control group (sh-NC). In the ESM1 group (sh-ESM1), 4.0 μg of ESM1 shRNA plasmid and 10 μL of transfection reagent were added to each well. Adjust the final volume to 2 mL/well with serum-free DMEM medium. The next experiment was carried out 48 h after transfection.

## CCK-8 experiment

Cells grown in logarithmic phase were seeded in three 96-well plates at $2\times10^5$ cells per well. After culturing for 24 hours, transfect according to the concentration of siRNA according to the instructions of the transfection reagent. Three replicate holes were set up for each group. After culturing for 72 h, 10 μL of CCK-8 reagent was added to each well. Incubate in an incubator for 4 h and measure the absorbance at 450 nm with a microplate reader. Cell viability (%) = (siRNA group-blank group)/(control group-blank group).

## Scratch test

Culture Hela cells in six-well plates. $3\times10^5$ cells were planted in each well, and the group without any treatment was used as the control group. The group that was transfected according to the instructions of the transfection reagent was the experimental group. After each group of cells adhered to the wall, use a 10 μL pipette tip to draw 3 straight lines in each well. Aspirate the culture medium and rinse gently with PBS twice. Take pictures with an inverted microscope, add siRNA for transfection, and continue culturing. Photographs were taken with an inverted microscope at 24 hours.

## Transwell experiment

Culture Hela cells in a 24-well plate with $2\times10^5$ cells per well. Take no treatment group as the control group. The cells treated according to the instructions of the transfection reagent were the experimental group. Trypsin was routinely digested, and the cells were resuspended in serum-free medium at a concentration of $1\times10^6$/mL. Place the Matrigel glue in the Transwell chamber for 1 hour in the incubator. Add 200 μL of cell suspension to the upper chamber, and 600 μL of 10% FBS to the lower chamber. Cultured for 24h, washed with PBS the next day. Fix with methanol and rinse. After washing with crystal violet staining, photographs were taken under a microscope.

## qRT-PCR experiment

By this method, the RNA expression levels of cell-related genes were detected to evaluate the effect of ESM1 on cell-related factors. Cells were collected and total RNA was extracted using Trizol reagent following the instructions. RNA was reverse transcribed into cDNA using a reverse transcription kit. Design specific real-time PCR primers based on the human gene sequence of NCBI Genebank. The whole real-time PCR process was operated according to the instructions of SYBRR Premix Ex Taq TM II. After adding the sample, the tubes were mixed and centrifuged. Amplification was accomplished using Fast Start Universal SYBR Green Master (ROX). The reaction conditions were as follows: pre-denaturation at 95°C for 5 min; denaturation at 95°C for 30 s; annealing at 60°C for 30 s; extension at 72°C for 50 s; 40 cycles; extension at 72°C for 5 min, and data collection. Three replicate tubes were set up for each sample, and the experiment was repeated 3 times. The relative expression of mRNA in each sample was calculated by $2^{-\Delta\Delta CT}$ method.

## Western blotting

The total proteins in the cultured cells were isolated using the RIPA lysate buffer (KeyGen, China). Western blot analysis was performed using the primary antibodies ESM1, ZEB1, E-cadherin, Vimentin, and GAPDH (CST signaling, USA), followed by secondary antibody (CST signaling, USA). Blots were detected using the enhanced chemiluminescence detection kit (Millipore, USA).

## Subcutaneous xenograft tumor model

Twenty-four SPF-grade BALB/c (nu/nu) female nude mice, 5–6 weeks old, weighing 20-22g, were provided by the Charles River Experimental Animal Center (animal certificate number: SCXK (Beijing) 2012–0001). The size and quality of tumors formed by different groups of cervical cancer cells in nude mice were detected by this method. Evaluation of the effect of ESM1 on the tumorigenic ability of cervical cancer cells in vivo. Cells in logarithmic growth phase were injected into the axilla of mice. Prepare about 100μL medium containing $1\times10^6$ cells and mix well for inoculation. They were raised in SPF environment, and the growth status and tumor growth of nude mice were observed. The tumor size was measured with a caliper every 2 days after inoculation. The volume ($mm^3$) was calculated according to the standard formula: length $\times$ width$^2$/2. At the end of the experiment, the tumors were dissected, and the net tumor weight of each mouse was measured. This study was carried out in strict accordance with the recommendations in the Guide for the Care and Use of Laboratory Animals. The experimental protocol was approved by the Research Ethics Committee of the second hospital of Hebei Medical University with the approval number of 2023-AE001. When a near-death happens or the width reaches to 15 mm, it will be the humane endpoint, carbon dioxide was used for euthanasia.

## Immunohistochemistry

Formalin-fixed-paraffin-embedded tissue sections were routinely dewaxed, hydrated, and microwaved for antigen retrieval. The sections were blocked with 3% hydrogen peroxide and 10% goat serum, respectively. Prepare Ki67 primary antibody working solution with PBS solution at a ratio of 1:100. Cover tissue sections, shake and incubate overnight at 4˚C. After the unbound primary antibody was washed with PBS solution, horseradish peroxidase-labeled goat anti-rabbit secondary antibody was used to bind the corresponding primary antibody. DAB method showed positive protein. Compute section staining scores. The staining intensity of histochemical sections was evaluated using ImageJ software. The average gray value (staining intensity) and positive area percentage (staining area) of positive cells were used as IHC measurement indicators. No positively stained cells is 0 point, <25% is 1 point, 25–50% is 2 points, >50%-75% is 3 points, >75% is 4 points.

## Immunofluorescence

The slipper was placed on a 24-well plate and cells were inoculated on the slipper at a rate of $1\times104$ cells per well. After transfection, cells were fixed with 4% paraformaldehyde solution at room temperature for 30min. The cells were rinsed 3 times with PBS for 5min each time. After rinsing, cells were permeated on ice for 10min with cell permeation solution (PBS solution of 0.5%Triton-100+1%NGS). The cells were closed at room temperature for 30min with 1%NGS PBS solution. After closure, the antibody was diluted at a ratio of 1: 100. The cells were incubated overnight at 4˚C with the diluted antibody solution. Finally, fluorescent secondary antibody and DAPI were used to stain the cells in turn under the condition of avoiding light. After staining, the cells were observed under a fluorescence microscope and photographed.

## Molecular docking

Search the protein crystal complex of ESM1 in the PDB database, process the target protein with the help of Schrodinger software, and remove water molecules, phosphate radicals and redundant inactive ligands in the target protein at the same time. In the Schrodinger software, hydrogenation, charge and other operations were performed on the target protein after

treatment. Import the 2D structure of aloe-emodin into Schrodinger software, and save its 3D structure in SDF format. The 3D structures of the target protein and aloe-emodin were uniformly set into a format recognizable by Schrodinger software. Identification of active pockets of ESM1 target proteins. Molecular docking of the target protein and aloe-emodin was carried out, and the affinity between aloe-emodin and ESM1 was compared and analyzed.

## Statistical analysis

Statistical analysis of data was performed using SPSS 17.0, and the results were expressed as mean ± standard deviation. The mean comparison between two groups of samples was used by t test. One-way analysis of variance was used to compare the means among multiple groups of samples. P<0.05 means the difference is statistically significant.

## Results

### 1. The expression and clinical significance of ESM1 in cervical cancer tissues

As shown in Fig 1A, the subcellular expression of ESM1 is mainly located in the cytoplasm. Immunohistochemical results showed that ESM1 staining intensity was higher in cervical cancer than in healthy tissue. The analysis of ESM1 expression in cervical cancer and paracancer tissues showed that the average expression level of ESM1 in cervical cancer tissues was higher than that in corresponding paracancer tissues (Fig 1B and 1C). Using the median expression value of ESM1 in cervical cancer tissue as cut-off point, 292 patients with cervical cancer were divided into ESM1 high expression group (n = 146) and ESM1 low expression group (n = 146). Kaplan-Meier analysis showed that cervical cancer patients with high ESM1 expression had shorter overall survival (Fig 1D). We further analyzed the expression of ESM1 by single cell sequencing. Fig 1E shows the expression clustering and correlation results of ESM1. The results showed that ESM1, a part of the 14 endothelial cell cluster, is mainly expressed in endothelial cells and is associated with angiogenesis. Fig 1F shows the expression of ESM1 cell line. The results showed that ESM1 affected endothelial cells and fibroblasts, we hypothesized that its function might be related to cervical cancer migration and EMT [21].

### 2. Functional enrichment analysis of ESM1 and its co-expressed genes in cervical cancer

To investigate the role of ESM1, we first analyzed the genes positively correlated with the expression of ESM1. Fig 2A shows the cluster analysis results of ESM1 co-expressed genes. ESM1 and its interacting proteins were further analyzed. As shown in Fig 2B, each target participating in the interaction is represented by a circle, and the larger the area of the circle, the greater its Degree. The thicker the circumference of the circle, the greater its Betweenness. The protein-protein interaction network results showed that ESM1 has a wide range of protein interaction networks. Subsequently, the GO functional enrichment of ESM1 and its co-expressed genes in the occurrence and development of cervical cancer was analyzed. GO items were screened out based on P-value<0.01, including Biology Process, Molecular Function, and Cellular Component. The results of functional enrichment analysis showed that biological processes mainly involved: Extracellular matrix organization, Angiogenesis, Cell adhesion, Regulation of epithelial cell migration, etc. (Fig 2C). The molecular functions mainly involve: Extracellular matrix structural constituent, Extracellular matrix binding, Extracellular matrix structural constituent conferring tensile strength, etc. (Fig 2D). Cell composition mainly involves: Basement membrane, Cell junction, Focal adhesion, etc. (Fig 2E). It shows that ESM1

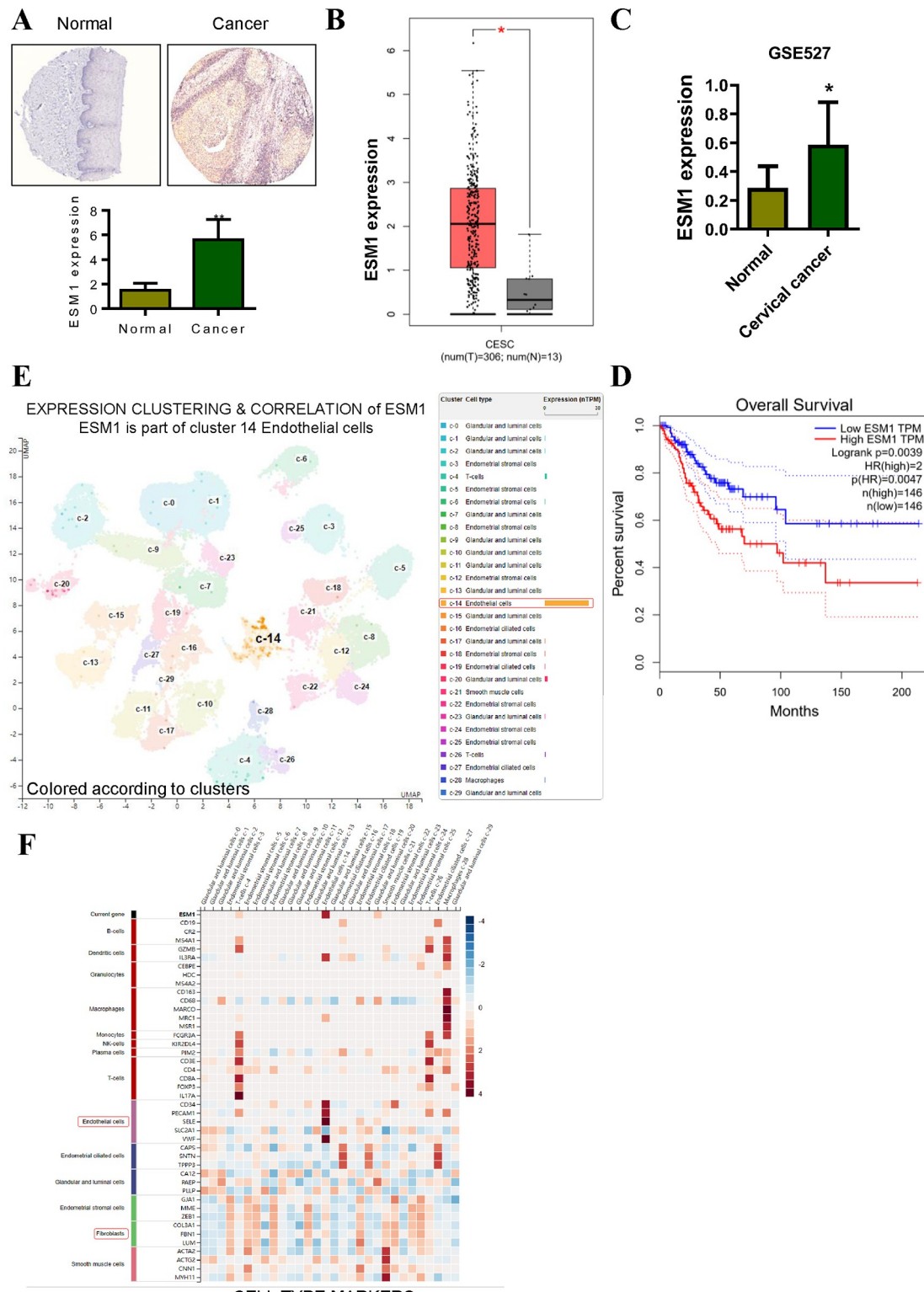

**Fig 1. ESM1 is significantly upregulated in cervical cancer and is associated with poor prognosis. A.** Statistical analysis of immunohistochemical results showed that ESM1 was highly expressed in cervical cancer. **B.** TCGA big data analysis shows that ESM1 is upregulated in cervical cancer. **C.** The GEO data analysis shows that ESM1 is upregulated in cervical cancer. **D.** Survival curves of patients stratified by ESM1 expression. **E.** Experssion clustering and correlation of ESM1 in single cell sequencing results. **F.** Correlation analysis between ESM1 and markers of different cell types. *P < 0.05, **P < 0.01 by two-tailed Student's t-test.

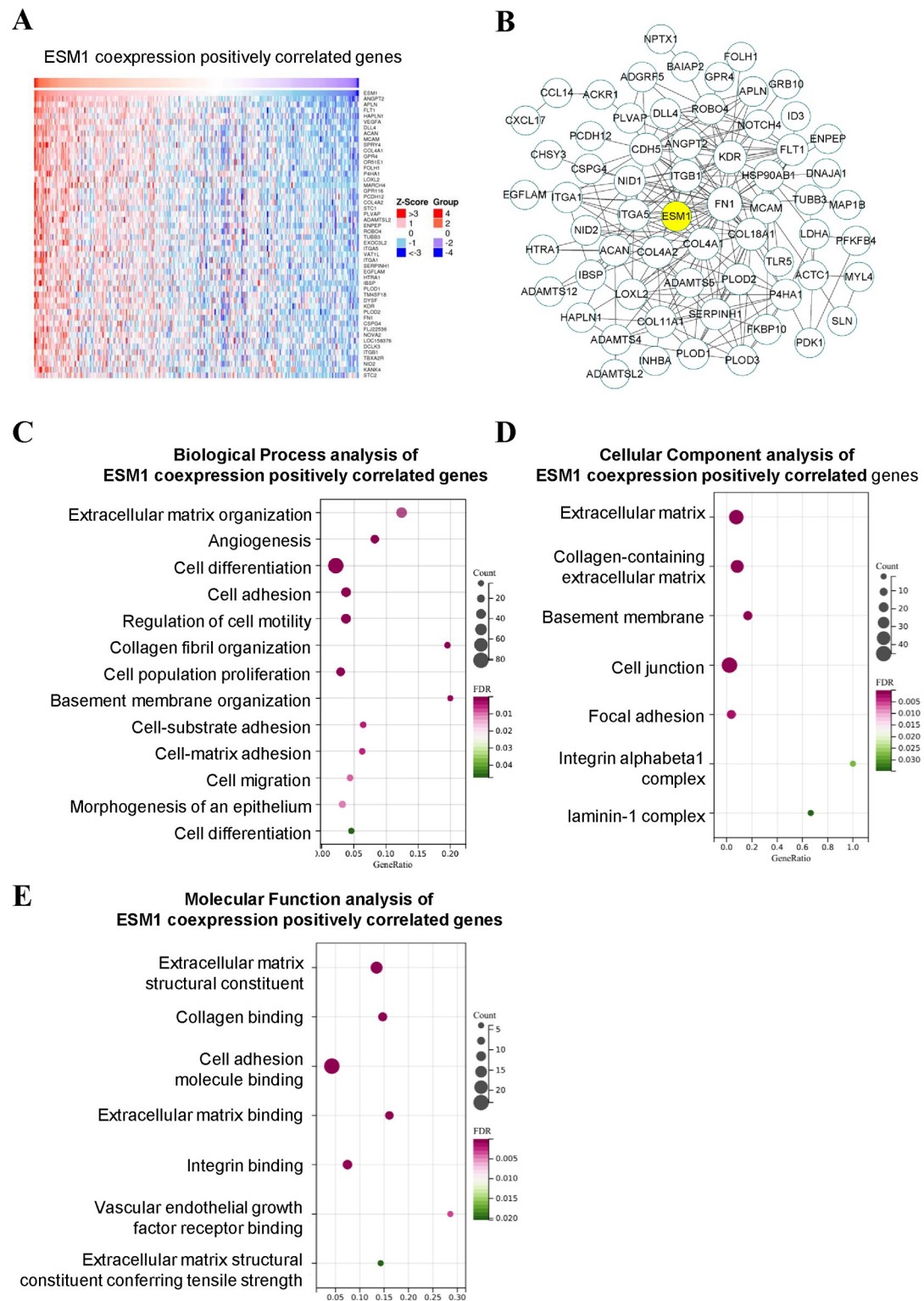

**Fig 2. ESM1 co-expression gene analysis, ESM1 affects EMT-related molecular mechanisms. A.** Cluster analysis of ESM1 and its co-expressed positively correlated genes in cervical cancer. **B.** PPI network of ESM1 and its co-expressed positively correlated genes. **C–E.** GO functional enrichment of ESM1 and its co-expression positively correlated genes.

may play a role in promoting cervical cancer by regulating extracellular matrix remodeling, migration and metastasis.

## 3. ESM1 promotes the proliferation, migration, invasion and EMT of cervical cancer cells in vitro

Based on the results of bioinformatics analysis, we further confirmed the role of ESM1 and its effect on EMT of tumor cells through in vitro experiments. In order to explore the specific effects of ESM1 on cervical cancer cells, we first detected the expression level of ESM1 in different human cervical cancer cell lines. Results showed that ESM1 was highly expressed in cervical cancer cell lines (Fig 3A). RT-qPCR results showed that the mRNA expression of ESM1 in Hela cells was reduced after siRNA treatment, indicating that siRNA successfully interfered with the expression of ESM1. After transfection of SiHa cells with the overexpressed plasmid, the level expression of ESM1 mRNA in SiHa cells in the vector-ESM1 group was up-regulated compared with the control group, and the difference was statistically significant (Fig 3B). Cell count results showed that after 72h of cell inoculation, the number of cells in the si-ESM1 group was significantly lower than that in the corresponding control group. In SiHa cells, the number of cells in the vector-ESM1 group was significantly higher than that in the vector-NC group (Fig 3C). The cell scratch test results showed that, compared with the control group, the migration ability of Hela cells was significantly inhibited after 24h siRNA treatment, and the difference was statistically significant (P < 0.01). After overexpression of ESM1, the migration ability of SiHa cells was enhanced (Fig 3D). Transwell test confirmed that after 24h of siRNA treatment, Hela cell invasion ability of si-ESM1 group was reduced compared with si-NC group, with statistical significance (Fig 3E). By interfering the expression of ESM1 with siRNA, we found that ESM1 acts as an oncogene in cervical cancer, promoting the proliferation, invasion and migration of cervical cancer cells. Further, we detected the expression of EMT-related markers E-cadherin and Vimentin. The results showed that overexpression of ESM1 could inhibit the expression of E-cadherin and promote the expression of Vimentin. However, silencing ESM1 can inhibit EMT (Fig 3F and 3G). The western blot results showed that when ESM1 was knocked down, ZEB1 and Vimentin were also down regulated, while E-cadherin was upregulated; when ESM1 was overexpressed in SiHa cells, the protein expressions of ZEB1 and Vimentin were increased, while the E-cadherin's expression was decreased (Fig 3H). The results of immunofluorescence were consistent with those of qRT-PCR and western blotting. Overexpression of ESM1 promotes EMT in SiHa cells (Fig 3I). Exogenous overexpression of ESM1 significantly enhanced the proliferation, invasion and migration of cervical cancer cells.

## 4. ZEB1 is the downstream target of ESM1

In order to further explore the molecular mechanism of ESM1, we analyzed the downstream regulation mechanism of ESM1. Through experiments, we found that ESM1 can affect the expression of ZEB1, and thus affect the changes of EMT-related indicators. SiHa cells confirmed that overexpression of ESM1 promoted the expression of ZEB1. Hela cell results showed that silencing ESM1 decreased ZEB1 expression (Fig 4A). In addition, ESM1 overexpression promoted MMP-2 and MMP-9 expression. Hela cell results showed that silencing ESM1 decreased the expression of MMP-2 and MMP-9 (Fig 4B and 4C). Co-expression correlation analysis results showed that ESM1 and ZEB1 had positive co-expression correlation in cervical cancer tumor tissues (Fig 4D). These results suggest that ESM1 can promote EMT process of cervical cancer cells through ZEB1.

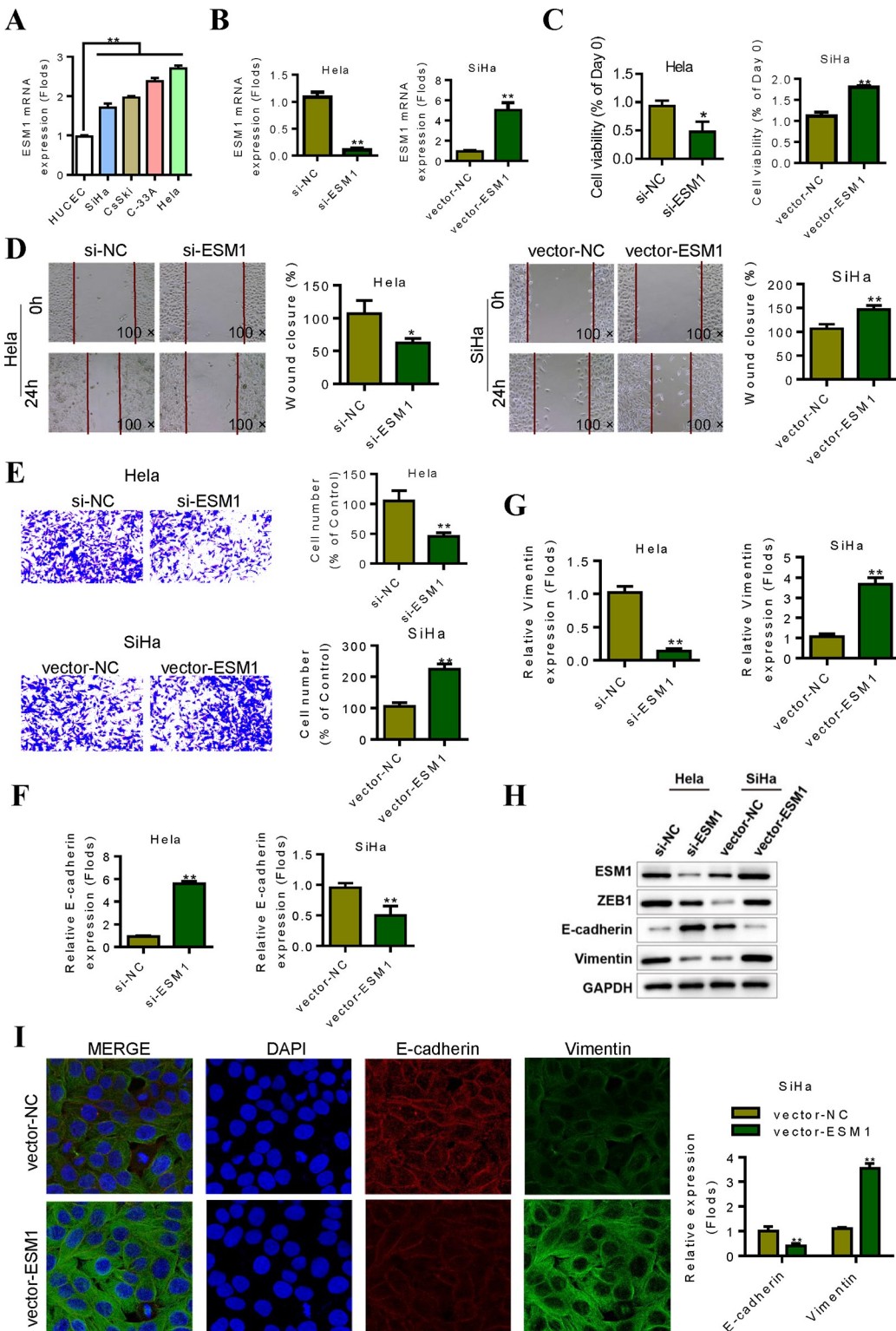

**Fig 3. ESM1 promotes the proliferation, migration and invasion of cervical cancer cells in vitro. A.** qRT-PCR analysis of the expression of ESM1 in human normal cervical epithelial cells (HUCEC cell line) and cervical cancer cells (Hela, SiHa, C-33A and CsSki). **B.** Detection of Hela cells transfected with siRNA ESM1 by qRT-PCR. qRT-PCR was used to detect the expression of ESM1 in SiHa cells transfected with ESM1 plasmid and control cells. **C.** CCK-8 analysis shows that downregulation of ESM1 inhibits DNA synthesis in Hela cells. Overexpression of ESM1 promotes DNA synthesis in SiHa

cells. **D.** Cell scratch analysis showed that the downregulation of ESM1 inhibited the migration ability of Hela cells. Cell scratch analysis showed that overexpression of ESM1 promoted the migration ability of SiHa cells. **E.** Transwell experiment, the downregulation of ESM1 inhibited the invasion ability of Hela cells. Transwell experiment, overexpression of ESM1 promoted the invasion ability of SiHa cells. **F-G.** The expression levels of EMT-related markers were detected in Hela and SiHa cells after different treatments. **H.** The western blotting results showed that protein expression levels of EMT related markers were regulated by ESM1. **I.** The expression levels of EMT related markers E-cadherin and Vimentin were detected by immunofluorescence. *P < 0.05, **P < 0.01 by two-tailed Student's t-test.

## 5. Effect of silencing ESM1 on tumor formation of cervical cancer cells in vivo

Tumor formation experiment results in nude mice showed that the tumor formed in the Hela-shESM1 group with 24d back implantation was significantly smaller than that in the Hela-shNC group (Fig 5A), and the tumor volume and tumor weight were also significantly smaller

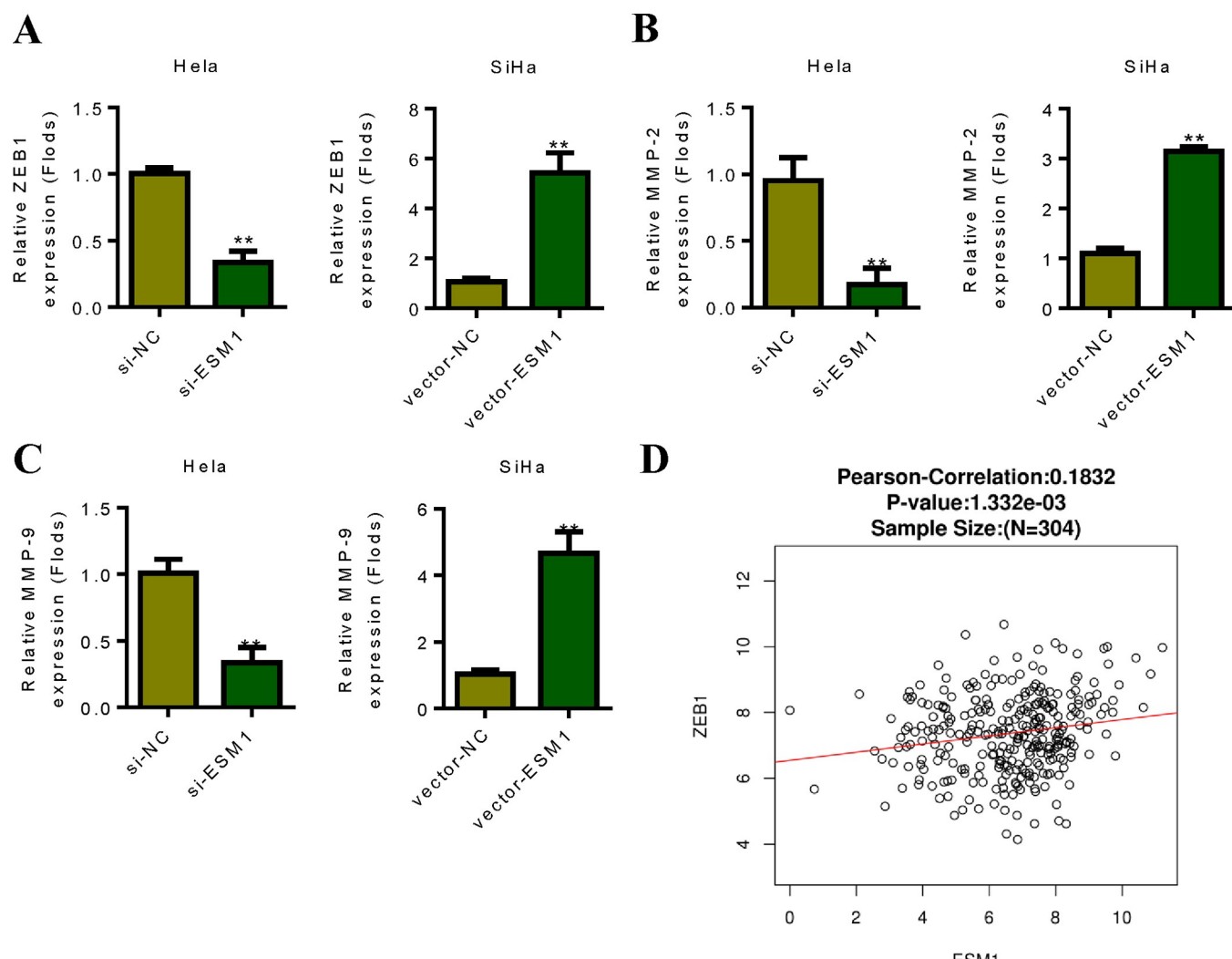

**Fig 4. ZEB1 is a downstream target of ESM1. A.** SiHa cells confirmed that overexpression of ESM1 promoted the expression of ZEB1, and the results of Hela cells showed that silencing ESM1 decreased the expression of ZEB1. **B.** SiHa cells confirmed that overexpression of ESM1 promoted the expression of MMP-2, and the results of Hela cells showed that silencing ESM1 decreased the expression of MMP-2. **C.** SiHa cells confirmed that overexpression of ESM1 promoted the expression of MMP-9, and the results of Hela cells showed that silencing ESM1 decreased the expression of MMP-9. **D.** Correlation detection of co-expression of ESM1 and ZEB1. *P < 0.05, **P < 0.01 by two-tailed Student's t-test.

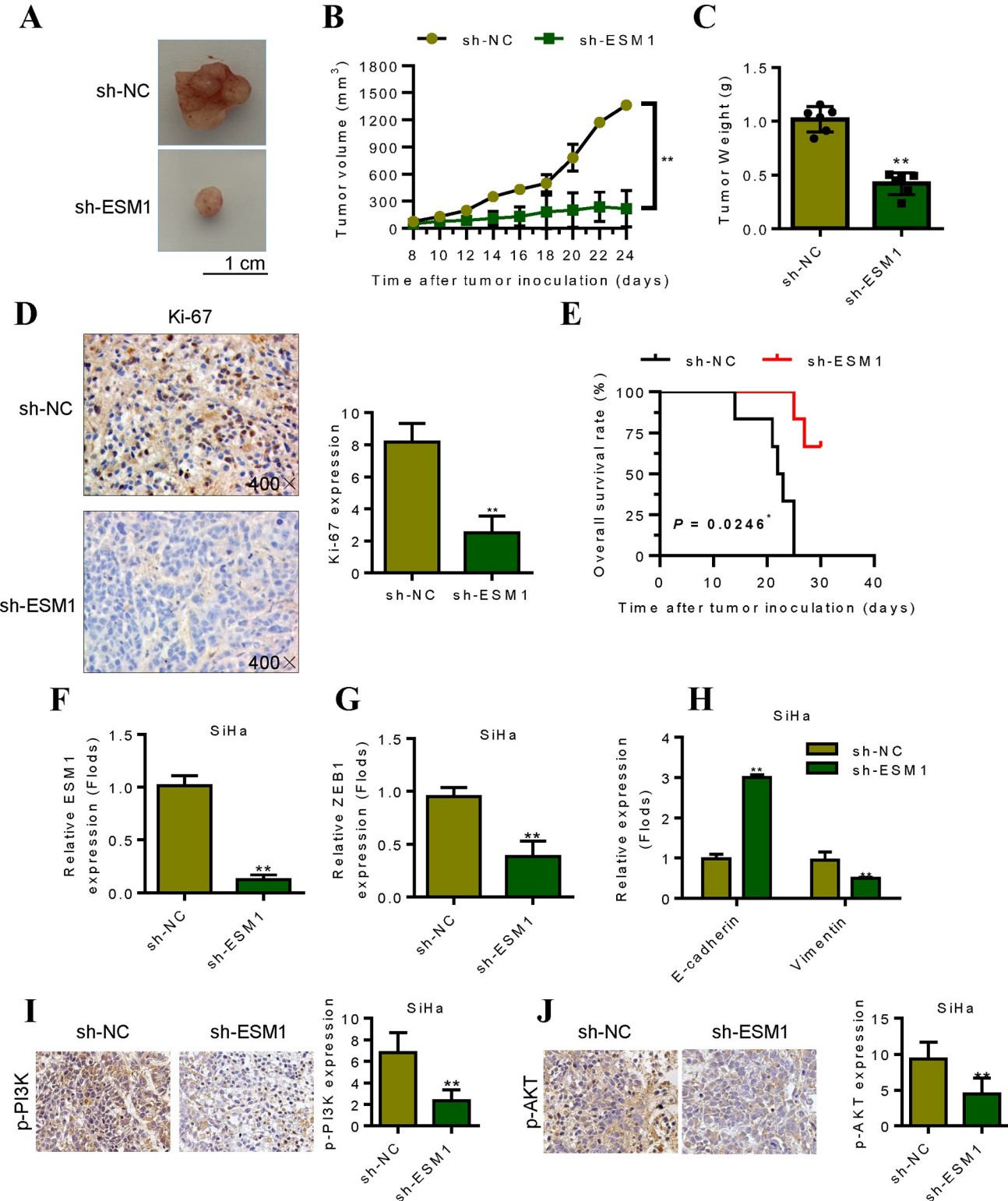

**Fig 5. ESM1 promotes the growth and metastasis of cervical cancer cells in vivo. A.** Experimental picture of subcutaneous xenograft tumor. **B.** Tumor volume graph. **C.** Tumor weight detection. **D.** Immunohistochemical detection of Ki-67 expression. **E.** Survival analysis of mice after tail vein injection of cervical cancer tumor cells. **F-G.** The expression levels of ESM1 and ZEB1 were detected in tumor tissues after different treatments. **H.** The expression levels of E-cadherin and vimentin were detected in tumor tissues after different treatments. **I-J.** The expression levels of p-PI3K and p-AKT were detected in tumor tissues after different treatments. *P < 0.05, **P < 0.01 by two-tailed Student's t-test.

than that in the Hela-shNC group (P<0.01, Fig 5B and 5C). The maximum diameter of the tumor size in this study was about 14 mm. Immunohistochemical test results showed that Ki-67 protein staining was strong in the tumor tissue of blank control group, while Ki-67 protein staining intensity was significantly decreased in ESM1 interference group (Fig 5D). Tumor cells were injected into the tail vein to construct the metastasis model. The results showed that compared with the Hela-shNC group, the survival of nude mice in the Hela-shESM1 treatment group was prolonged (Fig 5E). qRT-PCR confirmed that silencing of ESM1 decreased the expression of ESM1 and ZEB1 in tumor tissues (Fig 5F and 5G). Subsequently, we detected EMT-related indicators, including the expression changes of E-cadherin and Vimentin. Results showed that silencing of ESM1 up-regulated the expression of E-cadherin. Contrary to the E-cadherin results, overexpression of ESM1 promoted the expression of Vimentin, while silencing of ESM1 decreased the expression of Vimentin (Fig 5H). In addition, tumor results confirmed that ESM1 silencing reduced the p-PI3K/p-AKT expression (Fig 5I and 5J). These results suggest that silencing ESM1 expression can inhibit tumor formation of cervical cancer cells *in vivo*.

## 6. High-throughput drug screening showed that AE could bind to ESM1

Based on the discovered ESM1 function, we hypothesized that drug molecules capable of inhibiting ESM1 should have anticancer activity. Therefore, we screened active molecules from the traditional Chinese medicine monomer library through high-throughput drug screening. Fig 6A shows the process of high-throughput drug screening. Through Basic Filter 1 (Lipinski's "Rule of Five"), Glide docking (standard precision) and Glide docking (extra precision) are used for screening. The results showed that AE had the best binding activity with ESM1. The amino acid residues of AE interacting with ESM1 are shown in Fig 6B. Fig 6C shows the result of three-dimensional interaction between AE and ESM1.

## 7. Inhibitory effect of AE on ESM1/ZEB1/EMT signaling pathway and cervical cancer cells

Cell proliferation experiment results showed that the proliferation activity of Hela and SiHa cells decreased after AE treatment (Fig 7A). In order to observe the effect of AE on the invasion ability of different cervical cancer cell lines, we conducted Transwell invasion assay. Transwell test confirmed that after 24h treatment with AE, cell invasion ability of the AE treatment group was reduced compared with Hela and SiHa control groups, with statistical significance (Fig 7B). In order to explore the specific effects of AE on cervical cancer cells, human cervical cancer cell lines (Hela cells and SiHa cells) were administered. qRT-PCR detection showed that the expressions of ESM1 and ZEB1 in the cells treated with AE were significantly lower than those in the control group (Fig 7C and 7D). Subsequently, we detected EMT-related indicators, including the expression changes of E-cadherin and Vimentin. Results confirmed that AE treated can decrease the expression of Vimentin, while up-regulated the expression of E-cadherin (Fig 7E). The results of tumor formation experiment in nude mice showed that the tumor volume formed by AE treated group was significantly smaller than that in control group (P<0.01, Fig 7F and 7G). Similarly, the tumor mass formed in the AE treated group was significantly lower than that in the control group (P<0.01, Fig 7H). The expression levels of ESM1 and ZEB1 in tumor tissues of AE treated group were decreased by qRT-PCR (Fig 7I and 7J). EMT marker detection results showed that the expression of E-cadherin was up-regulated in the AE treatment group, while the expression of Vimentin was down-regulated (Fig 7K and 7L). These results suggest that AE can inhibit the malignant evolution of cervical cancer by inhibiting ESM1/ZEB1/EMT mechanism.

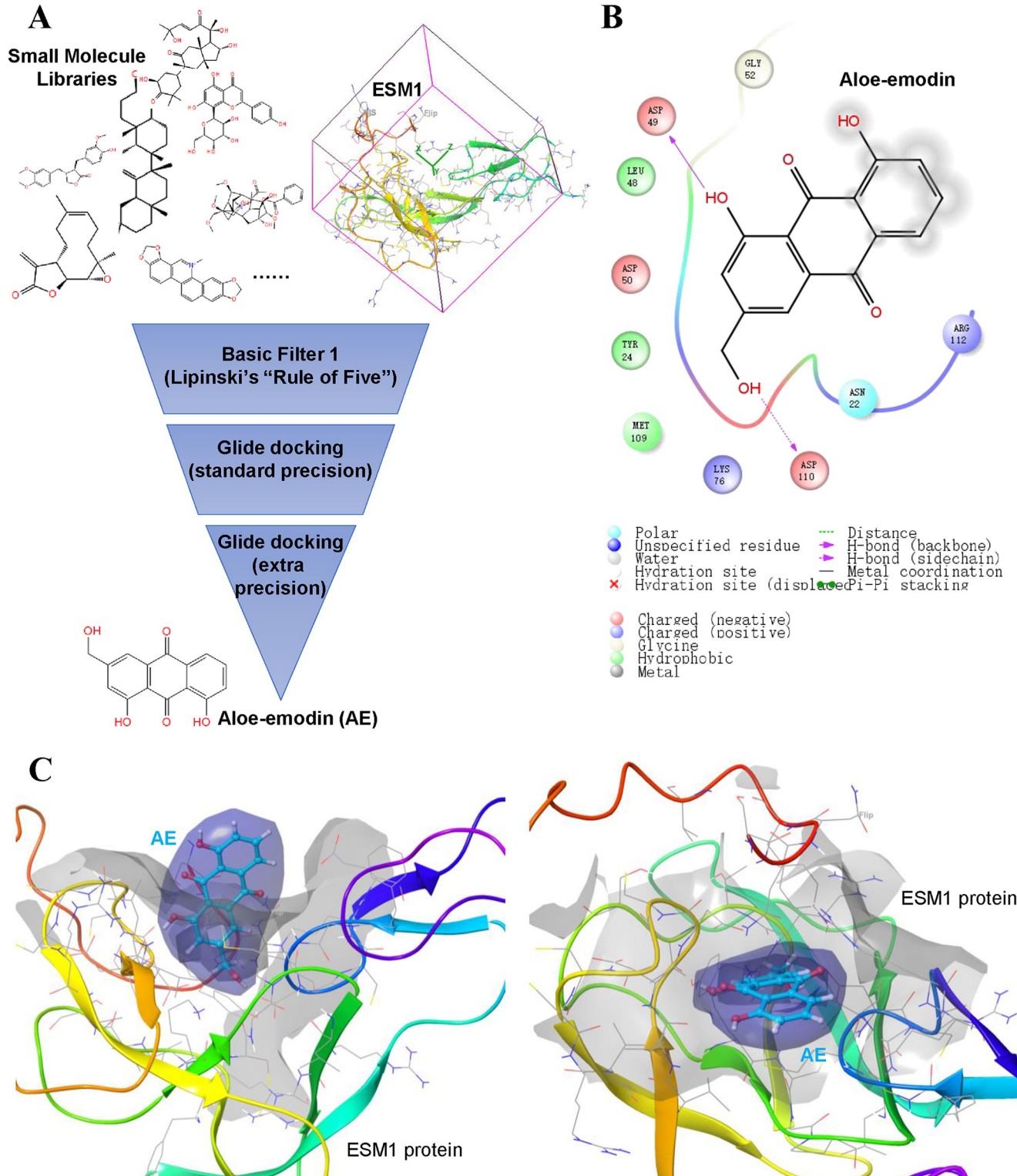

**Fig 6. High-throughput drug screening found that aloe-emodin can target ESM1. A.** The screening process of traditional Chinese medicine monomers based on the structure of ESM1 protein. **B.** aloe-emodin has the highest affinity with ESM1, a schematic diagram of the amino acid residues that they interact with aloe-emodin. **C.** Schematic diagram of the three-position interaction between aloe-emodin and the active pocket of ESM1.

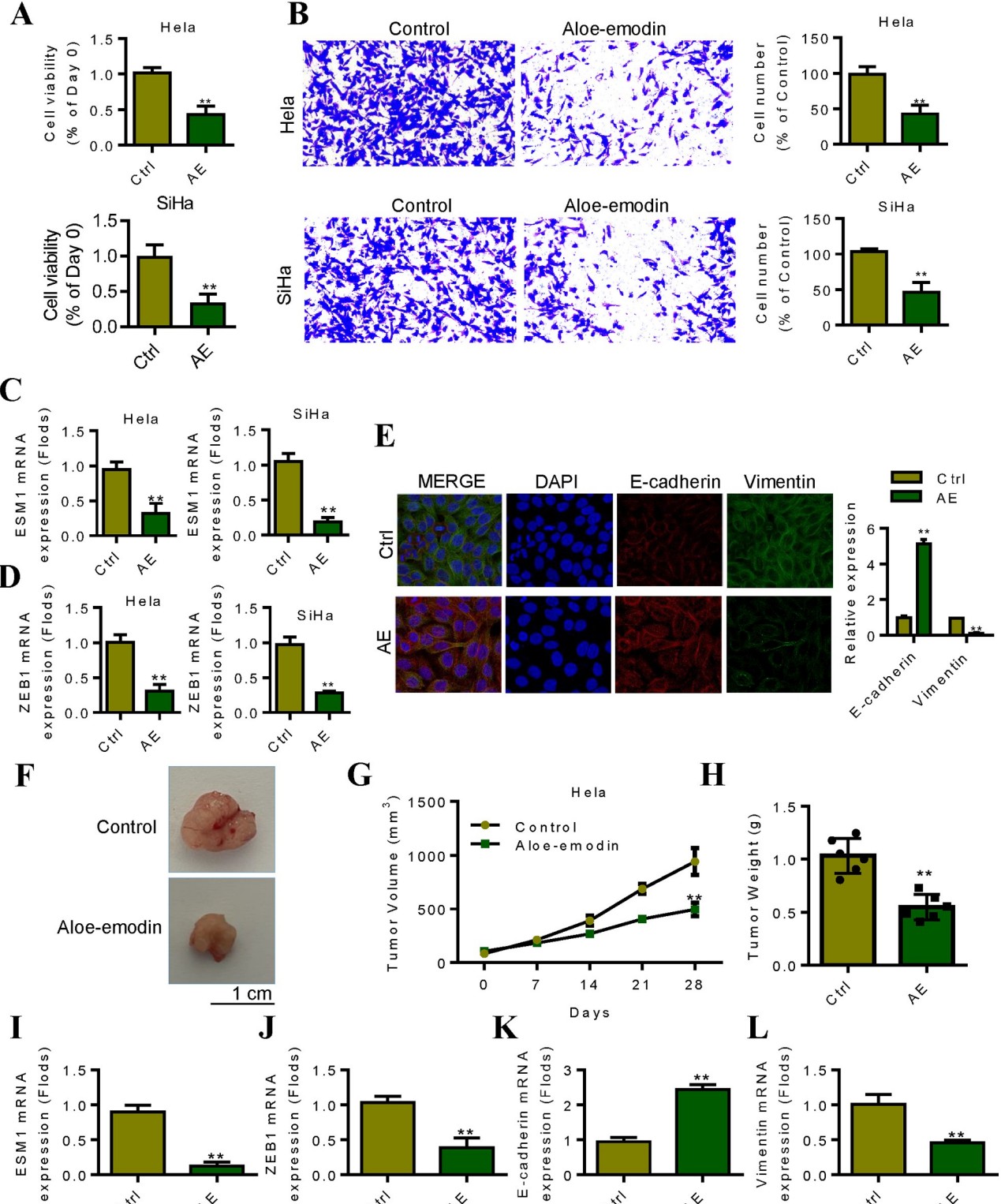

**Fig 7. Research on the effect of aloe-emodin on targeting and inhibiting ESM1/ZEB1/EMT signaling pathway to inhibit cervical cancer cells. A.** Study on the ability of aloe-emodin to inhibit the proliferation of cervical cancer Hela and SiHa cell lines. **B.** Study on the ability of aloe-emodin to inhibit the invasion of cervical cancer Hela and SiHa cell lines. **C.** qRT-PCR detection of aloe-emodin inhibiting the expression of ESM1 in cervical cancer Hela and SiHa cells. **D.** qRT-PCR detection of aloe-emodin inhibiting the expression of ZEB1 in cervical cancer Hela and SiHa cells. **E.** The expression levels of EMT related markers E-cadherin and Vimentin were detected by immunofluorescence. **F-G.** Detection of tumor volume of SiHa

cells after aloe-emodin administration. **H.** Detecting the tumor weight of SiHa cells after aloe-emodin administration. **I.** Detecting the expression of ESM1 in the tumor tissue after the SiHa cells were tumor-bearing and the aloe-emodin was administered. **J.** Detecting the expression of ZEB1 in the tumor tissue after SiHa cells were tumor-bearing and aloe-emodin administered. **K.** Detect the expression level of E-cadherin in the tumor tissue after the SiHa cell tumor-bearing and the administration of aloe-emodin. **L.** Detecting the expression of Vimentin in the tumor tissue after the SiHa cells were tumor-bearing and administered with aloe-emodin. *P < 0.05, **P < 0.01 by two-tailed Student's t-test.

## Discussion

Cervical cancer is a serious threat among female tumor patients, human papillomavirus (HPV), some sexually transmittable infections, and smoking are main causes of cervical cancer, although the cervical cancer could be partially prevented for highly effective primary (HPV vaccine) and secondary (screening) prevention measures, it still be a serious public problem among women [1, 3]. Advanced cervical cancer with tumor metastasis is the main reason for poor clinical prognosis of cervical cancer [22]. Thus, a comprehensive understanding of the mechanism underlying the process of metastasis is necessary to optimize the clinical treatment strategies for cervical cancer. A series of studies had demonstrated that EMT could mediate the metastasis and malignant progression of cervical cancer [23], during the EMT process, the downregulation of epithelial marker E-cadherin and upregulation of mesenchymal marker vimentin are key characteristics, which provides the tumor cells potential of metastasis [24]. As the expressions of E-cadherin and vimentin could be transcriptionally regulated by the EMT-activator ZEB1 [25], we could clarify the mechanism related to EMT through analyzing correlations with ZEB1/EMT axis.

In this study, the bioinformatics analyses showed that the ESM1 protein/gene were significantly upregulated in cervical cancer compared to the adjacent tissues, high ESM1 contents were significantly correlated with a poor clinical prognosis of cervical cancer patients; further experiment results demonstrated that ESM1 could promote the cervical cancer cell proliferation, metastasis, and invasion. To further clarify the mechanism of ESM1 in cervical cancer, omics analysis was conducted. The results showed that the co-expressed proteins with ESM1 could widely affect the cell adhesion, extracellular matrix organization, and mesenchymal cell differentiation, which indicated that ESM1 was likely to regulate the malignant progression of cervical cancer through EMT process. To verify the hypothesis, we further elucidated the stimulations of ESM1 to EMT in cervical cancer. Studies showed that upregulation of ESM1 could significantly promote the expression of EMT-activator ZEB1, the PI3K/AKT signaling pathway downstream of ZEB1 [26] was significantly activated, the transcriptionally regulated proteins of E-cadherin and vimentin were also significantly affected; when ESM1 was knocked down, the ZEB1 expression was significantly decreased, and the EMT related genes were correspondingly reversely regulated. These results verified that ESM1 could promote metastasis and malignant progression of cervical cancer through ZEB1/EMT axis, ESM1 was an important regulatory gene in cervical cancer, thus, ESM1 could be a potential target for further drug screening.

Grigoriu et al. [27] found that serum ESM-1 concentration in lung cancer patients was correlated with tumor type and stage, patient prognosis and tumor progression. ESM1 can guide the treatment of lung cancer to a certain extent. Similarly, Leroy et al. [28] found that serum ESM-1 concentration in patients with renal clear cell carcinoma can be used as an effective monitoring indicator for postoperative treatment and anti-angiogenic therapy in patients with renal clear cell carcinoma. Some scholars have found that the expression of ESM-1 in rectal cancer specimens is lower than that in normal rectal mucosa 5. It is currently believed that ESM-1 can regulate tumors through tumor-related inflammation, angiogenesis, lymphangiogenesis, and tumor cells themselves. It is also believed that compared with other normal tissues

with obvious hyperplasia, the high expression of ESM-1 in tumors has a strong specificity [10, 29, 30]. Malignancy is often accompanied by invasion and metastasis, which is also the reason why it is difficult to cure [31]. The occurrence of invasion and metastasis is related to a variety of complex mechanisms and processes [32]. Effectively inhibiting the invasion and metastasis of tumor cells is the starting point of many clinical and scientific treatments for cancer. It has been found that the matrix metalloproteinase family (MMPs) [33, 34], especially matrix metalloproteinase-2 (MMP-2) and matrix metalloproteinase-9 (MMP-9), as common factors promoting invasion and metastasis, play a regulatory role in the development of many tumors. This study also found that silencing ESM1 could reduce the expression of MMP-2 and MMP-9.

Studies have found that aloe emodin can inhibit the proliferation and metastasis of tumor cells by affecting a variety of signaling pathways. In breast cancer cells, aloe emodin down-regulates ERα-associated protein levels by inhibiting estrogen receptor alpha (ERα) transcription. In addition, it can inhibit the binding of ERα and heat shock protein 90 (HSP90) in cytoplasm, thus achieving the effect of inhibiting the proliferation of cancer cells [35]. Abnormal activation of the protein kinase (Akt)/Extracellular regulatory protein kinase (ERK) signaling pathway is common in various cancers and is closely related to the differentiation and proliferation of cancer cells. Aloe emodin has been shown to act on esophageal cancer cells activated by the Akt/ERK pathway in a dose-dependent manner, inhibiting Akt and ERK phosphorylation and significantly reducing the number of S-phase cells [36]. In this study, high-throughput drug screening was conducted to screen the potential anti-cervical cancer drug based on ESM1, finally, aloe-emodin was identified to be a potential targeted drug of ESM1. Aloe-emodin is an anthraquinone found in rhubarb, aloe, and other Chinese herbs [37], it was proved to have effects on anti-inflammatory, antiviral activity, and anti-cancer [38–40]. In this study, we found that aloe-emodin could effectively target the active pocket of ESM1, which meant that aloe-emodin might have potential treatment value in cervical cancer. Then, the in vitro and in vivo experiments were conducted to verify this hypothesis. The results showed that aloe-emodin could effectively inhibit the proliferation, invasion of cervical cancer, further results demonstrated that aloe-emodin inhibited the ESM1/ZEB1/EMT axis.

## Conclusion

In summary, we demonstrated that ESM1 was correlated with the malignant progression of cervical cancer through ZEB1/EMT axis, and ESM1 was a key regulatory protein during the cervical cancer progression. As ZEB1 was a transcription factor, we think this pathway should play a key role, and it will be a strong complement to the knowledge of ESM1 in regulating cervical cancer except angiogenesis pathway and PI3K-Akt/EMT pathway [41, 42]. Meanwhile, we further verified that the Aloe-emodin could directly target ESM1 to inhibit malignant progression of cervical cancer, which illustrated that aloe-emodin might be a potential drug and provided potential therapeutic strategy for clinical treatment of cervical cancer (Fig 8). However, we do think some limitations exist in this research, further studies are needed to provide a more comprehensive understanding of the therapeutic potential of AE and firmly establish the validity of these conclusions, which including: collaborating with clinical partners to access a larger cohort of patient samples to reinforce our findings; conducting long-term treatment studies to observe the prolonged effects and potential resistance mechanisms of cervical

**Fig 8. Proposed regulatory mechanism of ESM1 in regulating cervical cancer malignant progression.**

cancer; further investigations into the molecular mechanisms by which AE modulates ESM1 and its downstream signaling pathways.

## Author Contributions

**Conceptualization:** Sufen Zhao.

**Data curation:** Xiaoyan Zhu.

**Formal analysis:** Xiaoyan Zhu.

**Project administration:** Jie Qi, Jie Li.

**Writing – original draft:** Sufen Zhao.

**Writing – review & editing:** Sufen Zhao.

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
