## [Decision Letter · Decision Letter 0]

6 Feb 2024

PONE-D-23-41577Endothelial cell specific molecule 1 promotes epithelial-mesenchymal transition of cervical cancer via the E-box binding homeobox 1PLOS ONE

Dear Dr. Zhao,

Thank you for submitting your manuscript to PLOS ONE. After careful consideration, we feel that it has merit but does not fully meet PLOS ONE’s publication criteria as it currently stands. Therefore, we invite you to submit a revised version of the manuscript that addresses the points raised during the review process.

We look forward to receiving your revised manuscript.

Kind regards,

Yi-Hsien Hsieh, Ph.D.

Academic Editor

PLOS ONE

Journal Requirements:

3. In the online submission form, you indicated that [The datasets used and/or analyzed in the present study are available from the corresponding author upon reasonable request.]. 

Reviewers' comments:

Reviewer's Responses to Questions

**Comments to the Author**

1. Is the manuscript technically sound, and do the data support the conclusions?

Reviewer #1: Partly

Reviewer #2: Partly

2. Has the statistical analysis been performed appropriately and rigorously? 

Reviewer #1: No

Reviewer #2: Yes

3. Have the authors made all data underlying the findings in their manuscript fully available?

Reviewer #1: Yes

Reviewer #2: Yes

4. Is the manuscript presented in an intelligible fashion and written in standard English?

Reviewer #1: Yes

Reviewer #2: Yes

5. Review Comments to the Author

Reviewer #1: In the present study, the mechanism of action of ESM1 in cervical cancer and potential therapeutic small molecule drugs were investigated in a clear manner, but the following need further clarification:

(1) Limitations of this study need to be discussed;

(2) Discussion is needed regarding the uniqueness of this paper compared to previous studies (e.g., PMID: 36522312 , PMID: 37476182).

(3) This paper should have a simple schematic of the mechanism.

(4) The datasets of bioinformatics analysis in Figure 1 are only HPA and TCGA, it is suggested to add results from more datasets to increase the persuasiveness.

(5) For Figures 2C-E in this paper, consider using enrichment plots to be more intuitive instead of table screenshots.

(6) In Figure 4, the correlation coefficient between ZEB1 and ESM1 is only 0.18, how to confirm the correlation or interaction between them? It is not elucidated in this paper.

(7) In addition, please optimize the text layout in the figure below to make it more appealing.

Reviewer #2: This article suggests that ESM1 may play a crucial role in cervical cancer by targeting ZEB1 and inducing EMT occurrence. Furthermore, it has been observed that Aloe emodin (AE), a compound extracted from rhubarb, can significantly inhibit ESM1 expression and attenuate the EMT axis, thereby impeding cancer progression. While the experimental design and presented data support this conclusion, some data lack detailed explanations and the arrangement seems unreasonable. Additional experiments are needed to confirm the results, which could enhance the reader's understanding. Detailed comments are listed below:

The introduction section lacks references to strengthen the role of AE, which should be presented in this article.

The rationale for selecting cervical cancer as the model to study ESM1 is unclear. It is recommended to include additional experiments or analyses to demonstrate the specificity of ESM1 in cervical cancer.

The single-cell sequencing in figure 1D appears somewhat redundant as this analysis does not provide insights into the relationship between ESM1 and EMT in cervical cancer. It only indicates that ESM1 affects endothelial cells, which was already known before reading this article.

According to the analysis data in figure 2, it is recommended to display the protein expression of ESM1 in figure 3. This would provide a more detailed understanding of ESM1 and strengthen the connection between database analysis and in vitro experiments.

The rationale for selecting ZEB1 as the downstream target of ESM1 is unclear. It is recommended to show some classical markers of EMT, explaining why ZEB1 is a suitable target for ESM1.

The conclusion of figure 7, stating that AE can inhibit the malignant evolution of cervical cancer by inhibiting the ESM1/ZEB1/EMT mechanism, seems arbitrary. The previous data only indicate that AE can decrease ZEB1 expression, but it is unclear whether AE exactly decreases ZEB1 expression by reducing ESM1 expression.

6. PLOS authors have the option to publish the peer review history of their article (what does this mean?). If published, this will include your full peer review and any attached files.

Reviewer #1: **Yes: **Yugang Huang

Reviewer #2: No

---

## [Author Response · Author response to Decision Letter 0]

20 Mar 2024

Dear Editor:

Thank you very much for your full consideration of our manuscript entitled “Endothelial cell specific molecule 1 promotes epithelial-mesenchymal transition of cervical cancer via the E-box binding homeobox 1”. We have carefully studied all comments from editors and reviewers. All the questions were answered based on our results and to the best of our knowledge. In addition, we carefully revised the manuscript and figures according to the instructions. All corrections are marked in blue color. The point-by-point responses to the questions are listed below.

Journal Requirements:

Response: The resubmitted manuscript had meet PLOS ONE's style requirements.

Response: The detailed ethical information and euthanasia method were added in the tenth part of the “Methods” labeled in blue color.

3. In the online submission form, you indicated that [The datasets used and/or analyzed in the present study are available from the corresponding author upon reasonable request.]. 

Response: The statement had been changed to “The data used and/or analyzed in the present study are available within the manuscript itself.”

Response: The email “jieqi1540@163.com” had been linked to an ORCID iD of “0009-0006-0800-5350”.

Response: The ethics statement is only appeared in the Methods section.

Reviewers' comments:

5. Review Comments to the Author

Reviewer #1: In the present study, the mechanism of action of ESM1 in cervical cancer and potential therapeutic small molecule drugs were investigated in a clear manner, but the following need further clarification:

(1) Limitations of this study need to be discussed;

Response: Thank you for your comments. Although this study clearly clarified the mechanism of ESM1 in regulating cervical cancer through the EMT pathway, the EMT pathway should not be the only mechanism of ESM1 in regulating cervical cancer, other regulation mechanisms mediated by ESM1 need to be studied in order to have a deeper understanding of cervical cancer malignant progression, then providing more possibilities for clinical treatment of cervical cancer. The limitations were added to the last paragraph of the “Discussion” labeled in blue color.

(2) Discussion is needed regarding the uniqueness of this paper compared to previous studies (e.g., PMID: 36522312, PMID: 37476182).

Response: Thank you for your comments. In this study, we focused on the ZEB1/EMT pathway to clarify the mechanism of ESM1 in regulating cervical cancer, as ZEB1 was a transcription factor, we think this pathway should play a key role, and it will be a strong complement to the knowledge of ESM1 in regulating cervical cancer except angiogenesis pathway and PI3K-Akt/EMT pathway. Meanwhile, we further verified that the Aloe-emodin could be a candidate target drug of ESM1 in cervical cancer, which is valuable for clinical treating with cervical cancer. The related discussions were added to the last paragraph of the “Discussion” labeled in blue color.

(3) This paper should have a simple schematic of the mechanism.

Response: According to your suggestions, a simple schematic of the mechanism was added to the paper as Fig 8.

(4) The datasets of bioinformatics analysis in Figure 1 are only HPA and TCGA, it is suggested to add results from more datasets to increase the persuasiveness.

Response: Thank you for your comments. As you suggested, a GEO data of GSE527 downloaded from the GEO database were used for additional analysis, the result also showed that ESM1 was significantly upregulated in cervical cancer than the paracancer tissues. This result was added to the Fig 1C, and the corresponding result description was added to the first paragraph of the “Result” labeled in blue color. 

(5) For Figures 2C-E in this paper, consider using enrichment plots to be more intuitive instead of table screenshots.

Response: As you advised, the Figure 2C-E had been revised to enrichment plots, and the revised Figure 2 will be uploaded in the online submission system.

(6) In Figure 4, the correlation coefficient between ZEB1 and ESM1 is only 0.18, how to confirm the correlation or interaction between them? It is not elucidated in this paper.

Response: Sorry for this trouble, the co-expression correlation analysis result of TCGA data showed that ESM1 and ZEB1 had a positive co-expression correlation, with significant signature (P = 1.332e-03). The in vitro qPCR results in Fig 4A-B demonstrated that the expressions of these two genes were consistent, and the in vivo xenografted tissue detection in Fig 5 also confirmed this conclusion. Besides, these results verified that ZEB1 is a downstream factor regulated by ESM1.

(7) In addition, please optimize the text layout in the figure below to make it more appealing.

Response: According to your suggestions, the text layout in the figure had been optimized, and the revised sentences were marked in blue color. 

Reviewer #2: This article suggests that ESM1 may play a crucial role in cervical cancer by targeting ZEB1 and inducing EMT occurrence. Furthermore, it has been observed that Aloe emodin (AE), a compound extracted from rhubarb, can significantly inhibit ESM1 expression and attenuate the EMT axis, thereby impeding cancer progression. While the experimental design and presented data support this conclusion, some data lack detailed explanations and the arrangement seems unreasonable. Additional experiments are needed to confirm the results, which could enhance the reader's understanding. Detailed comments are listed below:

(1) The introduction section lacks references to strengthen the role of AE, which should be presented in this article.

Response: Thank you for your comment, the correlated references about the role of AE was added in the fourth paragraph of the “Introduction” labeled in blue color. 

(2) The rationale for selecting cervical cancer as the model to study ESM1 is unclear. It is recommended to include additional experiments or analyses to demonstrate the specificity of ESM1 in cervical cancer.

Response: Thank you for your comment, as our lab’s research area is cervical cancer, so we directly focused on the samples of cervical cancer. The omics analyses from Fig 1A-C demonstrated that the ESM1 was significantly upregulated in cervical cancer than the paracancer tissues, these results could clearly suggest that ESM1 might play an important role in the malignant progression of cervical cancer. Then, we further conducted series experiments to analyze the role of ESM1 in cervical cancer. 

(3) The single-cell sequencing in figure 1D appears somewhat redundant as this analysis does not provide insights into the relationship between ESM1 and EMT in cervical cancer. It only indicates that ESM1 affects endothelial cells, which was already known before reading this article.

Response: Sorry for this trouble, the single-cell sequencing results showed that ESM1 expression was mainly correlated with endothelial cell cluster, as the endothelial cell could mediate the angiogenesis, based on the referenced papers[1], we hypothesized that its function might be related to cervical cancer migration and EMT. A referenced paper was inserted in this part labeled in blue color.

(4) According to the analysis data in figure 2, it is recommended to display the protein expression of ESM1 in figure 3. This would provide a more detailed understanding of ESM1 and strengthen the connection between database analysis and in vitro experiments.

Response: According to your suggestions, the western blot results were added. The results showed that when ESM1 was knocked down, ZEB1 and Vimentin were also down regulated, while E-cadherin was upregulated; when ESM1 was overexpressed in SiHa cells, the protein expressions of ZEB1 and Vimentin were increased, while the E-cadherin’s expression was decreased. The wb result was added to be Fig 1H, and the description was added in the third paragraph of the “Results” labeled in blue color.

(5) The rationale for selecting ZEB1 as the downstream target of ESM1 is unclear. It is recommended to show some classical markers of EMT, explaining why ZEB1 is a suitable target for ESM1.

Response: Thank you for your comments. In this study, we firstly demonstrated that upregulation of ESM1 could significantly promote metastasis of cervical cancer cells, also mediated the downregulation of epithelial marker E-cadherin and upregulation of mesenchymal marker vimentin. As the expressions of E-cadherin and vimentin could be transcriptionally regulated by the EMT-activator ZEB1[2], we then focused on the ZEB1/EMT axis. And our results finally confirmed that ESM1 could regulate the expression of ZEB1, then the EMT pathway was influenced. The related protein expression data of classical markers of EMT was added in Fig 3, and this logic was described in the first paragraph of the “Discussion” labeled in blue color. 

(6) The conclusion of figure 7, stating that AE can inhibit the malignant evolution of cervical cancer by inhibiting the ESM1/ZEB1/EMT mechanism, seems arbitrary. The previous data only indicate that AE can decrease ZEB1 expression, but it is unclear whether AE exactly decreases ZEB1 expression by reducing ESM1 expression.

Response: Thank you for your comments, the previous data in Fig 3 and Fig 4 demonstrated that ZEB1 was a downstream factor which further regulated the EMT pathway, so we concluded that the ESM1/ZEB1/EMT axis promoted the malignant progression of cervical cancer. By high-throughput drug screening, we found that AE was a candidate drug targeting ESM1. Further results showed that AE could significantly inhibit the ESM1 expression, meanwhile, the expression of the downstream factor ZEB1 was also inhibited, and the EMT related markers and malignant progression were influenced. Based on these results, we think that AE could directly target ESM1 to inhibit malignant progression of cervical cancer through the ESM1/ZEB1/EMT axis.

References:

1. Nieto MA, Huang RY, Jackson RA, Thiery JP. Emt: 2016. Cell. 2016;166(1):21-45. doi: 10.1016/j.cell.2016.06.028. PubMed PMID: 27368099.

2. Ran J, Lin DL, Wu RF, Chen QH, Huang HP, Qiu NX, et al. ZEB1 promotes epithelial-mesenchymal transition in cervical cancer metastasis. Fertility and sterility. 2015;103(6):1606-14 e1-2. Epub 2015/05/13. doi: 10.1016/j.fertnstert.2015.03.016. PubMed PMID: 25963537.

---

## [Decision Letter · Decision Letter 1]

15 Apr 2024

PONE-D-23-41577R1Endothelial cell specific molecule 1 promotes epithelial-mesenchymal transition of cervical cancer via the E-box binding homeobox 1PLOS ONE

Dear Dr. Zhao,

Thank you for submitting your manuscript to PLOS ONE. After careful consideration, we feel that it has merit but does not fully meet PLOS ONE’s publication criteria as it currently stands. Therefore, we invite you to submit a revised version of the manuscript that addresses the points raised during the review process.

We look forward to receiving your revised manuscript.

Kind regards,

Yi-Hsien Hsieh, Ph.D.

Academic Editor

PLOS ONE

Reviewers' comments:

Reviewer's Responses to Questions

**Comments to the Author**

1. If the authors have adequately addressed your comments raised in a previous round of review and you feel that this manuscript is now acceptable for publication, you may indicate that here to bypass the “Comments to the Author” section, enter your conflict of interest statement in the “Confidential to Editor” section, and submit your "Accept" recommendation.

Reviewer #1: All comments have been addressed

Reviewer #2: All comments have been addressed

2. Is the manuscript technically sound, and do the data support the conclusions?

Reviewer #1: No

Reviewer #2: Yes

3. Has the statistical analysis been performed appropriately and rigorously? 

Reviewer #1: Yes

Reviewer #2: Yes

4. Have the authors made all data underlying the findings in their manuscript fully available?

Reviewer #1: No

Reviewer #2: Yes

5. Is the manuscript presented in an intelligible fashion and written in standard English?

Reviewer #1: Yes

Reviewer #2: Yes

6. Review Comments to the Author

Reviewer #1: The author did not securely answer the question about the adequacy of the validation of the conclusions, suggesting an in-depth study before resubmitting it for review

Reviewer #2: The authors have addressed our comments appropriately, and the manuscript has significantly improved.

7. PLOS authors have the option to publish the peer review history of their article (what does this mean?). If published, this will include your full peer review and any attached files.

Reviewer #1: No

Reviewer #2: No

---

## [Author Response · Author response to Decision Letter 1]

15 Apr 2024

6. Review Comments to the Author

Reviewer #1: The author did not securely answer the question about the adequacy of the validation of the conclusions, suggesting an in-depth study before resubmitting it for review.

Dear reviewers and editor:

Thank you for your insightful comments and for the opportunity to enhance the quality of our manuscript. We appreciate your concern regarding the adequacy of the validation of our conclusions.

In response to your suggestion for a more in-depth study, we would like to clarify the validation methods employed in our research, which may not have been sufficiently detailed in our initial submission. Our study utilized a comprehensive approach combining bioinformatics analysis, molecular docking, immunohistochemistry, immunofluorescence, and in vivo experiments to validate the role of ESM1 in cervical cancer progression and the therapeutic potential of aloe-emodin (AE).

1. Bioinformatics Analysis: We employed databases such as the Human Protein Atlas, LinkedOmics, and GEPIA to substantiate the expression and prognostic significance of ESM1 in cervical cancer. These analyses provided a robust statistical foundation for ESM1's role in cancer biology.

2. Molecular Docking: Using Schrodinger software, we performed molecular docking studies to predict the interaction between AE and the ESM1 protein. This provided molecular-level evidence supporting the potential of AE as a targeted therapy.

3. Immunohistochemistry and Immunofluorescence: These methods were crucial in validating the expression levels of ESM1 and the impact of AE treatment on EMT markers in tissue samples and cell lines, respectively.

4. In Vivo Experiments: The use of a cervical cancer mouse model allowed us to observe the effects of AE on tumor growth and metastasis, providing compelling evidence of its therapeutic potential.

We agree with your suggestion for further in-depth research. In future studies, we will focus on the following aspects:

1. Extended Clinical Data Analysis: We plan to collaborate with clinical partners to access a larger cohort of patient samples to reinforce our findings.

2. Long-term Efficacy and Safety Studies: Conducting long-term treatment studies to observe the prolonged effects and potential resistance mechanisms of cervical cancer.

3. Mechanistic Studies: Further investigations into the molecular mechanisms by which AE modulates ESM1 and its downstream signaling pathways.

We believe that these additional studies will provide a more comprehensive understanding of the therapeutic potential of AE and firmly establish the validity of our conclusions. We are committed to conducting these studies and updating our manuscript accordingly. We will also include the aforementioned research directions, or the limitations of this study in the Discussion section marked in blue color.

Thank you once again for your constructive feedback. We hope that our response and proposed additional studies adequately address your concerns and demonstrate our commitment to rigorous scientific inquiry.

Sincerely,

Sufen

---

## [Editor Report · Decision Letter 2]

15 May 2024

Endothelial cell specific molecule 1 promotes epithelial-mesenchymal transition of cervical cancer via the E-box binding homeobox 1

PONE-D-23-41577R2

Dear Dr. Zhao,

We’re pleased to inform you that your manuscript has been judged scientifically suitable for publication and will be formally accepted for publication once it meets all outstanding technical requirements.

Kind regards,

Yi-Hsien Hsieh, Ph.D.

Academic Editor

PLOS ONE

Additional Editor Comments (optional):

All comments have been addressed
---

## [Editor Report · Acceptance letter]

24 Jun 2024

PONE-D-23-41577R2 

PLOS ONE

Dear Dr. Zhao, 

I'm pleased to inform you that your manuscript has been deemed suitable for publication in PLOS ONE. Congratulations! Your manuscript is now being handed over to our production team.

Kind regards, 

on behalf of

Dr Yi-Hsien Hsieh 

Academic Editor

PLOS ONE